# Smart Contact Lenses in Ophthalmology: Innovations, Applications, and Future Prospects

**DOI:** 10.3390/mi15070856

**Published:** 2024-06-30

**Authors:** Kevin Y. Wu, Archan Dave, Marjorie Carbonneau, Simon D. Tran

**Affiliations:** 1Department of Surgery, Division of Ophthalmology, University of Sherbrooke, Sherbrooke, QC J1G 2E8, Canada; yang.wu@usherbrooke.ca (K.Y.W.);; 2Department of Medicine, University of British Columbia, Vancouver, BC V6T 1Z3, Canada; 3Faculty of Dental Medicine and Oral Health Sciences, McGill University, Montreal, QC H3A 1G1, Canada

**Keywords:** smart contact lens, ophthalmology, wearable electronics and sensors, clinical integration

## Abstract

Smart contact lenses represent a breakthrough in the intersection of medical science and innovative technology, offering transformative potential in ophthalmology. This review article delves into the technological underpinnings of smart contact lenses, emphasizing the current landscape and advancements in biosensors, power supply, biomaterials, and the transmission of ocular information. This review further applies new innovations to their emerging role in the diagnosis, monitoring, and management of various ocular conditions. Moreover, we explore the impact of technical innovations on the application of smart contact lenses in monitoring glaucoma, managing postoperative care, and dry eye syndrome, further elucidating the non-invasive nature of these devices in continuous ocular health monitoring. The therapeutic potential of smart contact lenses such as treatment through targeted drug delivery and the monitoring of inflammatory biomarkers is also highlighted. Despite promising advancements, the implementation of smart contact lenses faces technical, regulatory, and patient compliance challenges. This review synthesizes the recent advances to provide an outlook on the state of smart contact lens technology. Furthermore, we discuss future directions, focusing on potential technological enhancements and new applications within ophthalmology.

## 1. Introduction

With over two decades of research, recent advancements in multidisciplinary technology have paved the way for the development of smart contact lenses (SCLs). These devices, which survey ocular physiology and may respond in real time, fall within the domain of wearable monitoring devices. These devices consist of miniaturized technology which can record biomarkers and physical ocular properties while providing valuable information that may be used for diagnosis, monitoring, and treatment. The growing interest in SCL devices can be attributed to their multifaceted capacities. As such, SCLs can act in vision correction and be utilized as a non-invasive physiologic detector of health [1]. 

Beyond its role as a major sensory organ, the eye also functions as a gateway to the body’s overall health. Chemical and physical information discerned from the eye can be used to conduct indirect analyses on the health of the other organ systems. While there is a difference in the composition of tear fluid and blood, it is acceptable to use tear measurements as a representation of blood composition. Electrolytes such as glucose, Na+, K+, and Cl- can be measured through SCLs and translated into blood electrolyte levels. As such, healthcare providers can draw inferences about the health status of major organs and certain disease prognosis in patients [2] (Figure 1). SCLs may also be used as one critical component to help with making diagnoses of conditions such as diabetes. Using tear fluid analysis, tear glucose can be used to indirectly measure blood glucose. These values can be used to determine which patients have hyperglycemia and which require further monitoring of their hyperglycemia. Diabetic retinopathy, a serious consequence of high blood glucose in patients with long-term diabetes, is a major condition that may be prevented through continuous monitoring and management through SCLs.

Furthermore, SCLs can also be used to measure intraocular pressure (IOP), which is a major risk factor for the development of glaucoma. Glaucoma is a group of eye diseases characterized by the degeneration of the optic nerve and leads to progressive, irreversible damage to the optic disk and retinal ganglion cells. The diagnosis of glaucoma is based on IOP reading, cup-to-disk ratio, and visual field defects [3]. Due to the irreversible course of glaucoma, early detection is crucial, allowing for the healthcare team to respond promptly to changes in the patient’s condition. While individuals already diagnosed with glaucoma may certainly benefit from IOP sensing and the instant alerting capabilities of SCLs, those who belong to higher-risk populations of developing glaucoma may also find SCLs to be advantageous as well. Overall, this approach ensures a proactive therapeutic response and aids in the early detection of the disease.

SCLs may also play a role in relieving symptoms of dry eye syndrome, which include symptoms such as itchy eyes and burning or stinging sensations. Tear film instability, inflammation of the surface of the eye, and lacrimal glands are the key characteristics of dry eye syndrome and may be alleviated through the use of topical eye drops [4]. Conventional treatment may include artificial tears, topical steroids, antibiotics, or, in more serious cases, cyclosporine-A or lifitegrast drops [5]. SCLs may be beneficial in managing dry eye disease by serving as a drug delivery system to reduce ocular inflammation [6].

It is also no surprise that SCLs have the potential to revolutionize post-operative monitoring through the continuous supervision of vital signs such as IOP. Real-time data directly relayed from SCLs may provide guidance to clinicians in managing post-operative patients and can assist in detecting any deviations from normal values much earlier than standard post-operative check-ups.

This review will cover the basic technology accompanying SCLs and their current advancements. We will be commenting on the clinical applications of SCLs in conditions such as glaucoma, dry eye syndrome, post-operative management, and diabetes. In addition, we address areas of future research for SCLs. Furthermore, SCLs are heavily involved in the drug therapy and monitoring of various ocular diseases. While these wearable devices demonstrate encouraging outcomes, there are still regulatory limitations and technical challenges. In this comprehensive review article, we synthesize the recent literature to examine the role of SCLs in the diagnosis, monitoring, and management of ocular healthcare.

## 2. Smart Contact Lens Technology and Advancements

SCLs are different from regular contact lenses, primarily due to their advanced features that extend beyond basic vision correction. As previously introduced, SCLs are versatile tools for health monitoring due to embedded miniature electronic devices which enable additional functionalities such as analyzing changes in the composition of tear fluid for glucose, lactic acid, and other biomarkers to monitor various eye diseases [7]. However, it is crucial to strike a balance between user comfort and functionality. As such, enhancing the functionality of SCLs must not compromise their wearability and make them difficult to use. This entails the ingenious and careful assembly of components integrated into the SCL, ensuring that they do not add excessive weight, obstruct vision, or generate surplus heat. In addition, innovations in biomaterials also play a critical role, as researchers develop materials with suitable physical and chemical characteristics, which not only enhance SCLs’ capabilities, but also make them more comfortable for wearers.

This section will present the basic technology and biomaterials used in SCLs, focusing on novel innovations made in SCL design and biosensors while suggesting future considerations specific to ophthalmic applications. In general, SCLs are composed of various elements such as power supply, sensors, biomaterials, and circuits, which allow for the measurement and transmission of information to users or clinicians. The conventional applications of SCLs include the measurement of glucose and lactic acid concentration, IOP measurement, and drug delivery.

### 2.1. Measurement of Glucose and Lactic Acid

The measurement of glucose through miniaturized biosensors is limited to two main methods. One is the enzymatic degradation of glucose through glucose oxidase and the other is the use of non-enzymatic processes, which ultimately rely on physical changes in SCL to signify elevated glucose levels in the tear fluid.

Monitoring glucose levels is important for preventing ocular conditions like diabetic retinopathy, and accurate, sensitive biosensors play a key role in the diagnosis and management of such diseases. One of the preliminary glucose sensors in SCLs was developed by Badugu et al. (2003), which contained boronic acid containing fluorophores that were capable of binding to the glucose molecules present in tears. The subsequent reaction led to excitation and visual changes in fluorescence that could be measured and correlated with the glucose levels in tears. While this method set the precedence for non-invasive and continuous monitoring, there was much interference from other substances, and ensuring high accuracy was a challenge [8]. Since then, considerable progress has been made to increase glucose biosensor sensitivity and biocompatibility.

Park et al. (2024) recently developed an SCL glucose biosensor that was able to transmit information wirelessly to the user’s smartphone [9]. The group demonstrated an in vivo correlation between tear glucose and blood glucose. Glucose monitoring was facilitated by a working electrode and counter/reference electrodes that were embedded in the contact lenses and exposed to tear fluid. The working electrode was coated with glucose oxidase that was immobilized in a matrix composed of chitosan. Glucose oxidase converted glucose into gluconolactone and hydrogen peroxide. Hydrogen peroxide was reduced through the presence of Prussian blue, acting as an artificial peroxidase on the working electrode to release electrons, and a current was created that was proportional to the concentration of glucose in the tears (Figure 2). The group’s design was also equipped with near-field communication (NFC) capabilities to transmit glucose data in real time.

Additionally, Kim et al. (2022) demonstrated long-term continuous glucose monitoring using bimetallic nanocatalysts immobilized in nanoporous hydrogels [10]. The team proposed the use of gold platinum bimetallic nanocatalysts to resolve existing issues in glucose monitoring, such as the low stability of glucose oxidase and variable measurements. The authors also suggested that this novel innovation increases the sensitivity of glucose sensors and increases tear contact time with the biosensor, which produces a more accurate and consistent quantification of glucose content.

Interestingly, both studies from Park et al. (2024) and Kim et al. (2022) addressed challenges in the real-time monitoring of blood glucose [9,10]. While Park et al. (2024) proposed a concept of “personalized lag time” through continuous glucose monitoring, which may run into issues requiring extensive calibration for each user, Kim et al. (2022) introduced substantial advancements in material technology by using nanoporous hydrogels, which reduce hysteresis and lower the detection threshold while also enhancing sensitivity. However, the long-term stability of a nanocatalyst may be a concern, as there could be potential leaching or degradation that may result in shifts in baseline measurements. In addition, long-term evaluations for toxicity and inflammatory markers may be required. However, both studies demonstrated novel leaps in improving the glucose sensors within SCLs.

Han et al. (2023) developed modified hyaluronate gold platinum bimetallic electrodes for their construction of SCLs for continuous glucose monitoring [11]. The group suggested that the optimized design improved long-term stability, suppressed the crack formation typically found on conventional Au/Pt electrodes, and prevented gold dissolution by the chloride ions found in tears. The bimetallic electrodes are responsible for the release of electrons through the decomposition of hydrogen peroxide produced by the glucose oxidation reaction. Most notably, the group demonstrated a high correlation (ρ = 0.88) between the blood and tear glucose levels in rabbit models, supporting further advancements in clinical applications to monitor diabetes.

Guo et al. (2021) pursued another approach by using thin molybdenum disulfide (MoS_2_) nanosheets, which can sense the glucose concentrations in tear fluid [12]. Glucose oxidase is immobilized on the surface of the nanosheet and the electrons that are released are detected by MoS_2_ transistor. The group suggested that MoS_2_ nanosheets played a role in increasing the sensitivity of the biosensor due to their high surface to volume ratio and layered surface reducing the unspecific binding of other molecules in tear fluid, which could have inhibited glucose from diffusing towards the biosensor. Furthermore, the researchers demonstrated that the MoS_2_ nanosheets allowed for an excellent mechanical robustness and biocompatibility. While this design allowed for unobtrusive monitoring, the fabrication of the nanosheets may be complex and difficult to scale for commercialization.

Additionally, the quantification of glucose using sensors that utilize enzymes is prone to error due to being heavily influenced by environmental conditions such as temperature, pH, and other chemicals in the tear fluid which can lead to denaturation [13]. Lin et al. (2018) proposed an SCL using phenylboronic acid to absorb glucose and increase the thickness of the contact lens, which can be detected through the use of a smartphone [14]. While this method is safe from environmental pressures, the extent of contact lens expansion would need to be tightly correlated with glucose concentration to ensure that each measurement is reliable and accurate for clinical use. Ruan et al. (2017) demonstrated the use of a novel glucose sensor that utilized a crystalline colloid array which was embedded in a hydrogel matrix [15]. The lens emitted light waves from yellow, green, and blue spectra, which are associated with the glucose concentration in tear fluid. The group also proposed their novel glucose biosensor, which can detect very small (0.05 mM) concentrations of glucose and exhibits an excellent biocompatibility. However, similar problems exist with quantifying the exact glucose concentration in tear fluid, and it is uncertain if visual methods of measurement can be acceptable for diagnostic or monitoring purposes.

High lactic acid levels are indicative of oxygen deprivation and metabolic imbalances [16]. While lactic acid is typically measured in blood, using tear fluid to produce continuous data is a non-invasive approach that may be vital for patients who are at risk of ischemia. Thomas et al. (2012) developed a biosensor which incorporated lactate oxidase enzyme to recognize l-lactate. While lactate oxidase enzyme is known to be unstable, the authors immobilized the protein via cross-linkage with bovine serum albumin (BSA) and glutaraldehyde in order to mitigate this issue [17]. There is currently limited research on the use of lactic acid biosensors on SCLs. This may be due to a large focus on more common issues such as diabetes management or glaucoma diagnosis, which affect a larger population. SCLs which measure lactic acid production may target niche populations such as athletes to measure their physical performance.

### 2.2. Measurement of Intraocular Pressure (IOP)

IOP measurement is critical in the diagnosis and management of glaucoma. While IOP is measured using applanation tonometry by ophthalmologists [18], SCLs may have a role in the observation of glaucoma patients due to their pressure-measuring capabilities. Capacitive sensors are most used to measure IOP. A dielectric layer is placed between two electrodes, which conforms around the corneal curvature. The compression of the dielectric layer due to IOP leads to an increased capacitance. Zhu et al. (2022) developed a hydrogel-based SCL with a pyramid-microstructured capacitive sensor for wireless, real-time IOP monitoring. The authors encountered issues with hydrogel swelling, which was solved using a conformal stacking technique [19]. The group demonstrated an increased sensitivity of the capacitor sensor due to micropyramid structuring, hence increasing the accuracy of measurements in the IOP monitoring of in vitro porcine eyes, however, the fabrication process is notably intricate and may present large-scale production challenges. Yang et al. (2022) presents a wireless contact lens with a cantilever design of a capacitive sensing circuit [20]. This allowed for sensitive IOP monitoring and on-demand ocular drug delivery via iontophoresis. The group proposed that its cantilever design allows minimal interference between drug delivery circuitry, IOP monitoring, and vision. This design is an example of a novel strategy aimed at optimizing the limited space on SCLs and to maximize functionality without hindering vision and comfort. Similarly, Brobrowski et al. (2018) proposed hybrid circuitry which concurrently acted as biofuel cells and capacitors to ensure the full utilization of the limited space [21]. The device is a self-charging biosupercapacitor harnessing energy from tear glucose by incorporating indium tin oxide nanoparticles to produce electrode surfaces that enhance capacitive and catalytic currents. It would be interesting to add a capacitive sensor to this design to measure IOP and extensively test such devices in in vivo models.

Strain sensors are also used to measure IOP and work by detecting mechanical distortion within the lens material. The disruptions are transformed into electrical signals that are monitored [22]. These sensors are usually made from conductive and flexible materials such as metallic nanowires [23,24] and graphene [25,26]. Thin-film conductive polymers are emerging materials that should be investigated in the use of SCLs as strain sensors. These polymers are originally insulators which are changed to conductors through doping strategies [27]. The advantages of such polymers include an excellent electrical conductivity, transparency, flexibility, and great biocompatibility [27]. Strain sensors become optimal, that is, allow for immediate and accurate assessment, when they exhibit a high sensitivity and low detection limits. For example, Huang et al. (2022) developed a strain sensing module combined with a drug release module, composed of waterborne polyurethane and polycaprolactone-gelatin nanofibers that were enhanced by a gold layer. These modules have a detection limit of 0.1% strain and response time of 0.15 s, making them highly sensitive for monitoring physiological signals such as IOP and responding in real time [28]. Sensimed Triggerfish is an SCL sensor that utilizes strain gauges capable of detecting IOP changes over 24 h and has marketing approval in the USA. Hence, the device has demonstrated a satisfactory safety and tolerability during clinical trials [29]. This innovation significantly enhances the data and models that clinicals can use to identify IOP-related patterns, but further utility in being able to interpret and exercise the information gathered from the SCL is still the primary area of research.

Microfluid channel sensors are also used to detect IOP. Fluid within a network of microfluid channels responds to changes in IOP by corneal deformation. The fluid flows out of a sensing chamber and into a sensing channel as the chamber is compressed. The amount of fluid movement is calibrated to reflect specific pressure levels, which allows for the precise measurement of IOP. Fluid is often dyed and can be visualized on its own or through a smartphone, which can be used to quantify the amount of IOP. Yuan et al. recently constructed a novel microfluidic contact lens sensor that used dyed fish oil as an indicator liquid for tracking IOP changes [30]. The team proposed the addition of a bilateral wall to the sensor, which demonstrated an increase in overall sensitivity and linearity, allowing for precise and reliable IOP measurements. Another group utilized microfluid channel sensors to demonstrate autonomous drug delivery to the eye [31]. The design leveraged the mechanical pressure generated by blinking to ensure a steady and predictable flow of drugs through microchannels embedded into the SCL. As such, microfluid channel sensors are shown to be multi-purposeful in their utility. A key advantage of microfluidic channel sensors is their ability to operate without a power source, hence enabling true continuous monitoring. However, with any fluid passing through a channel, there may be risks of clogs, leaks, and subsequent irritation. Hence, An et al. (2018) conducted usability testing studies by performing blockage and leakage tests to ensure that the channels could withstand high pressures and were clear, as well as demonstrating control over the liquid in the channels for reliable drug delivery. The team’s silane coupling technique to bond plastic membranes with PDMS and subsequent novel thermoforming technique demonstrated microfluidic feasibility in SCLs by showing that the channel could be transformed into a curved shape to fit the eye without losing its structural integrity [32]. The group further demonstrated successful IOP monitoring on porcine eyes [33]. Agaoglu et al. (2018) manufactured a microfluidic-based sensor embedded within a silicone contact lens which successfully measured IOP on porcine eyes. The team was able to transform small strain changes into large detectable fluid volume changes that could be monitored using a smartphone camera. [34] While contact lens are an innovative way of measuring IOP, Araci et al. (2014) created an implantable microfluidic sensor that could be embedded within the intraocular lens of the eyes. The team proposed that the technology represents a promising step towards the better management of glaucoma by allowing more frequent and reliable measurements [35]. Similarly, Enders et al. (2020), proposed the Eyemate-IO system as an implantable medical device that directly measures IOP and relays the data to a smartphone. The authors demonstrated favorable safety and efficacy profiles, 12 months post-operatively [36]. However, the invasive nature of surgical procedures raises concerns about the long-term durability of the sensor, and potential inaccuracies in measurement due to conditions like hazy cornea may suggest SCLs to be a more viable option. Interestingly, Zhu et al. (2022) modified the use of microfluid technology to address dry eye from the use of regular contact lens [37]. The movement of eye blink and external eyelid pressure improves the tear fluid flow to keep the ocular surface well-hydrated. The capability of this innovation to reduce discomfort associated with prolonged contract lens wear supports its application in SCLs, especially for the extended monitoring of ocular conditions and relief from dry eye disease.

### 2.3. Power Supply

SCLs are fitted with complex circuity that requires electrical power for monitoring and the transmission of information. One such design of SCLs uses wireless transmission, such as Keum et al. (2020), who used a transmitter and receiver coil to power SCL for diabetic diagnosis and therapy [38], or Sensimed Triggerfish SCL, which includes an adhesive antenna attached to the skin around the orbit of the eye [29]. Power transmission through inductive coupling is conducted between a coil in the contact lens and in the adhesive antenna. which is powered externally through a battery that the patient wears. The external wear may also be equipped with a portable recorder of information that is transmitted from the contact lens sensor through Bluetooth connection [39]. The antenna generates a continuous magnetic field which powers the contact lens through induction.

Both RFID (radio-frequency identification) and NFC (near-field communication) are modes of power supply for SCLs. For example, Chiou et al. (2016) developed a wirelessly powered long-term IOP monitoring system for glaucoma patients using RFID for both communication and wireless power transfer. [40] The RFID tag is integrated into the contact lens and an RFID reader is positioned externally into eyeglasses. The main challenges faced by the authors included efficiently transferring enough power to operate the sensor continuously, particularly at the lowest possible level to avoid heating the eye and remain within the safety regulations set by IEEE (Institute of Electrical and Electronics Engineers) [41]. In addition, RFID tags must convert electromagnetic energy from the reader’s signal into electrical energy, which is prone to energy loss if the antenna design or impedance matching are inappropriate. The team used a loop antenna design, which was optimized to fit the limited space available on the contact lens and was required to avoid obstructing vision. Further modifications were also made, such as creating a unform magnetic field distribution to ensure consistent power transfer, adding a parallel capacitor to help tune the antenna, increasing the bandwidth to capture variations in transmitted signals, and matching the impedance to ensure that most radio frequency (RF) energy was absorbed by the antenna instead of reflected [40]. These design modifications are critical to maximize power transfer and must reflect a comprehensive strategy of both maximizing wearer comfort and the specific operational requirements of long-term medical monitoring.

Finally, the RF energy must be converted into direct current (DC) from alternating current (AC) to power the smart contact lens circuitry. This conversion is performed by a rectifier, commonly either through silicon or Schottky diodes. For example, Park et al. (2019) used silicon diodes to charge a supercapacitor within an SCL which required a stable DC input. The diodes were constructed using ultrathin silicon nanomembrane, which was crucial for maintaining the softness and flexibility of the contact lens and ensured it would provide comfort to the wearer and conform to the curvature of the eye. Most notably, Park et al. (2019) developed an SCL that integrated a rechargeable solid-state supercapacitor to enable reliable and continuous power supply to the electronic components, which is key for long-term, uninterrupted use. The team also ingeniously integrated the supercapacitor, antenna, rectifier, and LED in a compact form, leaving a small footprint on the SCL [42]. However, silicon diodes are known to have a high forward voltage drop and may be too large for ultra-low-power applications like SCLs [43]. Schottky diodes are also used as rectifiers and may offer lower forward voltage drops and fast switching capabilities [44]. Lower forward voltage drops allow less energy to be lost as heat, which is essential for SCLs, where power needs to be conserved and allocated efficiently. Takamatsu et al. (2019) utilized a Schottky diode in a wireless power transfer system for SCLs using a metal-air battery with zinc as the anode. The implementation of a Schottky diode was necessary, as the researchers addressed the potential issue of joule heating and ensured that the SCL circuitry did not adversely affect the ocular surface. The group developed a hybrid power battery which consisted of both a wireless power system and metal-air primary batteries. DC power was generated by zinc and bilirubin oxidase, while AC power was generated by wireless power. The group postulated that power boosting from both sources was necessary for a robust multifunctional battery system, as well as higher voltages and a better performance [45].

Solid-state batteries have been made to be thinner and more flexible in order to be implemented directly on curved polymer surfaces. Solid-state batteries provide a constant power supply, allowing for predictable and consistent usage, whereas wireless power transmission may lose connection if the device is moved away from the path of the energy source. Wireless power transmission also raises concerns about the continuous effects of exposure to electromagnetic waves to the eyes causing a safety issue and are also less efficient due to energy loss during transmission. In contrast, solid-state batteries have a high energy efficiency, but need to be physically integrated into the device and can negatively impact the comfort of the SCL. Lee et al. (2018) manufactured solid-state lithium-ion batteries which could be fabricated directly on hydrogel surfaces through a single-step annealing process [46]. However, a major concern is the leakage of toxic electrolytes from lithium-ion batteries. Yun et al. (2021) suggested using a sodium/potassium aqueous battery working in a tear-based solution [47]. By using tear solution as an electrolyte, there was a minimal risk of cytotoxicity. However, while the voltage produced was between 0.3 to 0.7 volts, which the team suggested was enough to operate low-power components, there may be higher voltage requirements if more modules are fitted in multipurpose SCLs. In contrast, solid-state batteries provide around three volts of power [46]. It is also important to consider that the novel development of processors and sensory modules may consume lower amounts of energy as they become more efficient due to the growing trend of miniaturizing larger components into nanoscale units.

Biofuel cells convert energy from biological substances into electricity. Kang et al. (2022) designed glucose fuel cells embedded within TRIS ([tris(trimethylsiloxy)silyl]propyl methacrylate) hydrogel using platinum nanowires as the anode and carbon nanotubes as the cathode to generate a stable power supply for SCLs [48]. Similarly, Falk et al. (2013) designed a biofuel cell using absorbate in tears rather than glucose to generate power [49]. While biofuel cells ensure self-powering capability without the need for an external power source, they only provide a voltage of around 0.5 volts, which may not be enough for multiple components installed on SCLs. Frei et al. (2018) proposed that 7 μW for 24 h would be adequate for many SCL applications. The team demonstrated that the output of Mg/air batteries was considerably higher at 40 μW for 24 h than abiotic glucose fuel cells at 2 μW for 24 h [50].

Novel methods of powering SCLs are also being investigated. For example, Lin et al. (2022) developed organic solar cells capable of providing the necessary power for biosensors that measure glucose and calcium ions. These solar cells, which are capable of absorbing photons and generating an electrical current, were optimized to operate efficiently under indoor lighting conditions and produced about 0.6 volts of power [51]. Triboelectric nanogenerators generate electricity from mechanical movements. Chu et al. (2016) and Pu et al. (2023) both proposed the use of this power source on a wide array of wearable electronics [52,53]. Triboelectric nanogenerators may have the potential to harness energy from the natural movements of the eyes such as blinking or normal eye movements, however, miniaturizing the manufacturing of such devices may be a challenge. Pourshaban et al. (2024) designed an SCL that uses both solar power and a metal-air power harvester that is powered through blinking movements [54]. As the eyelid moves across the harvester, the electrolytes in tear fluid spark an oxidation reaction and produces electrons that power the SCL. Yang et al. (2023) suggests the use of body heat to supply the electrical current from a thermoelectric generator. The team demonstrated this mode of energy harvesting through the production of a wearable bracelet that operated even with an extremely low power consumption [55]. A scaled-down design fit for SCLs may present a novel solution in generating power.

### 2.4. Biomaterials

Conventional SCL materials include a combination of conductive materials such as graphene [56], silver nanowires [23], or gallium-indium alloy [57] combined with hydrogels, such as polydimethylsiloxane (PDMS), silicone [38], poly(2-hydroxyethyl methacrylate) (PHEMA) [19], Poly (Ethylene Terephthalate) (PET) [58], Poly (Methyl Methacrylate) (PMMA) [59], and 2-Methacryloyloxyethyl Phosphorylcholine (MPC) [60]. PHEMA poly(2-hydroxyethyl methacrylate) hydrogel is a popular contact lens material, which is highly biocompatible due to its water/oxygen permeability and has excellent optical and hydrophilic properties [61]. PDMS is also commonly used due to its biocompatibility, optical transparency, and high oxygen permeability, making it soft and breathable. However, this often coated with PEGMA, which is shown to increase water content and glucose permeability [62,63].

Graphene is a distinctive, thin nanomaterial that demonstrates a high strength, electrical conductivity, transparency, and flexibility, while also exhibiting an excellent biocompatibility [59]. Hence, graphene is used in many sensor components. For example, Ku et al. (2020) utilized graphene in a field-effect transistor which could measure cortisol concentration and produce an electrical signal [57]. Furthermore, Lee et al. (2017) showed graphene coating on SCLs can protect against electromagnetic waves and protect the ocular surface from dehydration [64]. Since SCLs incorporate electronics that transmit electromagnetic signals, graphene could offer protective benefits against certain ocular diseases like glaucoma. Additionally, this material may also be used in SCLs that serve to combat dry eye diseases by ensuring minimal tear evaporation leading to the relief of symptoms. Graphene nanostructures were also demonstrated as being a cost-effective material through the use of drop-casting and then patterned using a direct laser interference patterning technique, potentially accelerating the commercialization of smart contact lenses while maintaining excellent optical and conductive properties. [56]

### 2.5. Transmission of Information

SCLs must transmit and receive information to effectively interact with external devices, such as smart phones and other medical monitoring devices. The enables real-time health monitoring and rapid therapeutic responses, which are crucial for patients with chronic conditions such as glaucoma or diabetes. Antennas are the mainstay for outgoing communication and their design is critical in maximizing size and preventing vision obstruction. Hence, a loop and spiral style antenna are promising design geometries for SCLs. A single loop antenna was developed by Chiou et al. (2017) and facilitated information transfer through RFID communication protocols [65]. Ting et al. (2014) also developed a single-loop antenna and measured the antenna radiation patterns, efficiency, and antenna gains when used on pig eyes. The team proposed their newly developed device for the wireless monitoring of eye health [66]. Kim et al. (2017) demonstrated the use of a spiral-style antenna made from a graphene-silver nanowire hybrid which was patterned lithographically on an ultra-thin parylene substrate [67]. Magnetic coupling was achieved between an external reader antenna and the sensor’s spiral antenna. The antenna supported not only communication transfer, but also enabled the device’s power needs. The spiral design allowed for both the measurement of the IOP by increasing capacitance and inductance through the expansion of the coils. This, in turn, changed the resonance frequency and was communicated to an external reader coil aligned over the sensor. Hence, while single antennas may be simpler and potentially cheaper to produce, spiral antennas are better suited for smart contact lenses as they can be designed to be extremely compact and have a wide range of frequences, which makes it easier to handle multiple functions such sensing and data transmission. Unlike single antennas, the omnidirectional radiation patterns of spiral antennas allow them to receive signals in different directions, which is important for SCLs, where the orientation of the reader may not be controllable. Similarly, Chen et al. (2017) decided to place spiral antennas in a distributed pattern, which also supported data and power transmission through near-field communication with a smartphone [68].

## 3. Therapeutic Potential

Without doubt, SCLs have the potential to transform the management and treatment of diseases such as dry eyes, glaucoma, and diabetes through ocular drug delivery and monitoring.

### 3.1. Dry eyes

Traditional methods like eye drops are inefficient with a low bioavailability due to drug loss from blinking and overflow from the eye [69]. Eye drops are also poorly distributed on the ocular surface and their effectiveness may also depend on patient compliance and technique [70]. On the other hand, SCLs may be engineered to deliver medication at a controlled rate and can be integrated with diagnostic tools to monitor treatment efficacy and health in real time, which can allow for adjustments as needed.

Targeted and on-demand drug delivery is a cornerstone of SCL development. Sun et al. (2024) developed an innovative contact lens that releases levofloxacin and diclofenac in response to elevated levels of reactive oxygen species (ROS) present at the site of ocular inflammation [71]. This model could be further developed not only to administer therapy, but also to monitor disease progression and treatment management. SCLs may detect levels of ROS and allow for the customization of drug release rates, leading to personalized medicine approaches in eye care. This is especially important in dry eye or post-operative applications where inflammation can widely occur on the ocular surface. On a broader scale, the maintenance of a consistent and accurate therapeutic level of antibiotic application directly where needed using SCL technology could help to lower the risk of antibiotic resistance and enhance compliance among users.

### 3.2. Glaucoma

Glaucoma patients can benefit from both the controlled application of medications to the eye and the accurate tracking of IOP through SCLs. Kumara et al. (2024) developed a contact lens for glaucoma management that can deliver a single drug (latanoprost) or two drugs simultaneously (latanoprost and timolol) to the ocular surface [72]. Their design relies on a chitosan-based nanocomposite material which entraps the drug product and is hydrolyzed in the presence of naturally occurring lysozyme in tear. The group demonstrated the significant release of drug product in the presence of lysozyme in vitro and no adverse effects on the rabbit eyes were indicated. Future studies should focus on integrating SCLs with sensors to monitor drug release and utilizing other biomarkers that could act as triggers to release drug products. This may direct and personalize the responsiveness of the SCL system to different stages of disease and ocular conditions.

Similarly, Xu et al. (2019) developed a contact lens that delivered two drugs, latanoprost and timolol, but they were encapsulated within micelles. The micelles were interwoven with the hydrogel matrix and released through the hydration and degradation processes of the polymer matrix [73]. The team proposed that this sustained release of drug products can be controlled through adjusting the cross-linking density, porosity, and hydrophilicity of the polymer.

Furthermore, Kim et al. (2022) manufactured an SCL with the on-demand delivery of timolol through the dissolution of gold channels. The team integrated an IOP sensor for the monitoring of glaucoma and demonstrated an excellent feasibility of drug delivery through SCLs by conducting immunohistochemical analysis to show significant differences between the treatment and control groups of glaucoma-induced rabbits. Ultimately, the team successfully incorporated an IOP sensor, drug delivery system, microchip, and wireless communication and power systems into the compact environment of a contact lens [74].

Without doubt, a key aspect of the therapeutic potential of SCLs is their capability to examine ocular disease progression through the measurement of physical properties such as IOP and chemical biomarkers in tear film. This monitoring helps ophthalmologists to assess the severity of the disease in real-time and adjust treatment programs accordingly. For example, matrix metalloproteinases (MMPs) are proteolytic enzymes that cleave extracellular matrix structures such as collagen and lead to structural remodeling. MMP is a biomarker of glaucoma and is found abundantly in early glaucoma. However, in severe stages, there is a notable reduction in MMP-9 activity in tears [75]. It may be possible to catch glaucoma early on and stage the disease accurately using SCLs with specific MMP biosensors. Ye et al. (2022) designed an SCL with the ability to measure both IOP and MMP-9 in tears [76]. The authors employed an anti-opal structure that changes colour in response to IOP and subsequently quantifies MMP-9 activity through surface-enhanced Raman scattering signals when specific substrates are cleaved. This design offers a non-invasive and direct approach to monitor the progression of glaucoma. Ongoing engineering improvements include integrating additional biosensors for other biomarkers without limiting the user’s vision or comfort and increasing the sensitivity and specificity of biosensors to enhance performance.

MMP-9 is also a biomarker for inflammation in many other ocular diseases. Jang et al. (2021) manufactured an SCL with a graphene field-effect transistor biosensor capable of measuring MMP-9 in tear fluid. The team expanded the design by including a heat patch on the upper and lower eyelids, which relieves the symptoms of meibomian gland dysfunction and is activated when MMP-9 is detected in the tear fluid [6]. This study provides both diagnostic function alongside instant rapid therapeutic action and lays the groundwork for the consistent and real-time treatment of ocular diseases. In addition, the quantification of inflammation from post-operative procedures may help clinicians to observe, manage, and mitigate recovery complications effectively.

### 3.3. Diabetes

SCLs can play an important role in both monitoring the glucose levels in tears, as well as administrating topical drugs to protect the eyes from retinopathies. While the finger-prink method to measure blood glucose levels is popular due its speed and simplicity, it can also lead to several complications such as infection [77], pain, and discomfort. Non-invasive methods for measuring glucose levels are the current focus of exploration and measuring the glucose in tear fluid is an emerging idea due to being strongly correlated with blood glucose levels [78]. Hence, the non-invasive nature of SCLs make them ideal for monitoring diabetes.

Keum et al. (2020) developed an SCL with a biosensor for monitoring the glucose levels in tears and included a flexible drug delivery system consisting of genistein to reduce the risk of severe diabetic retinopathy. The drug delivery system is controlled electrochemically through the dissolution of a gold membrane when a voltage is applied, and the subsequent drug product is released in a pulsatile matter. The system was tested in vitro and in vivo in rabbit models to assess the safety profile; the team demonstrated no significant temperature changes from the heat generated in operating the SCL. Most notably, the researchers showed comparable therapeutic effects of genistein delivered through SCL to that of Avastin through intravitreal injection [38]. This further exemplifies the non-invasive nature of SCLs and their potential to treat diseases without compromising clinical effectiveness

Seo et al. (2024) suggests the use of a colorimetric SCL to quantify the glucose levels in tear fluid, avoiding high manufacturing costs and the need for a power supply associated with electrochemical methods. The group applied a coating of poly(tannic) acid with boronic acid to the contact lens surface. This allowed for the reversible binding of glucose, which can be measured using a colorimetric assay. However, the lens must be removed to analyze the results, hence preventing the real-time monitoring of glucose levels [79].

Similarly, Jeon et al. (2021) developed a non-invasive optical monitoring system using a nanoparticle embedded contact lens that changed colour based on the glucose concentration in the tear fluid. The system was enhanced by image-processing algorithms and tested in vivo on mouse models and human tear samples. SCLs are trending towards optical systems whereby glucose levels can be visually quantified to avoid both the discomfort of traditional testing systems and reducing the bulky technology associated with electrochemical systems in SCLs.

## 4. Challenges and Limitations

Smart contact lenses offer a variety of potential uses in the field of ophthalmology. From diagnosing, managing, and treating disease, these non-invasive wearables may have the ability to change patients’ ocular health outcomes. However, despite their benefits, the implementation of SCLs still poses many challenges that need to be addressed.

The first challenge is ensuring that the SCL material and design are biocompatible, safe, and cause little irritation or adverse reactions when used long-term in the eye. There are few studies which demonstrate SCLs use for longer periods of time past 72 h [80,81]. Additionally, these studies use very small sample sizes and consist primarily of healthy individuals, therefore, the findings may not be generalizable to the sick population.

SCLs encounter challenges in ensuring the proper hygiene and maintenance of the lens, as well as patient compliance. In fact, a study by Cardona et al. (2021) revealed significant non-compliance with recommended practices, such as the proper storage and care of lenses [82]. These behaviors could similarly affect SCL usage, potentially leading to additional complications such as infection. Similarly, patients may not even wear prescriptive SCLs due to their comfortability or difficulties in handling the lens, which can lead to decreased adherence and potentially compromise their ocular health.

Additionally, current research on SCLs is still in its primitive stage. Most studies are limited to being in vitro and to rabbit or porcine ocular models. Unfortunately, the results obtained in animal studies cannot fully translate to human conditions, as both eye models differ in tear production rates and eye surface characteristics [83]. Also, short- and long-term user comfort can only be investigated in clinical trials and many investigational studies that test SCL use in vivo often do so in the short term, sometimes only for a couple of days. For that reason, more studies should explore the long-term effects of continuous use to determine biocompatibility and potential chronic side effects. Although significant research has been conducted, more clinical studies are necessary before they can be put on the market. Nonetheless, current SCL research demonstrates substantial innovation in both technology and monitoring capabilities, setting the stage for personalized medicine.

It is no surprise that the constant evolution of circuit miniaturization, alongside innovation in nanotechnology and 3D printing has sped up progress in SCL research. However, technical challenges are also abundant in the manufacturing of SCLs. Creative engineering solutions must be implemented to integrate multiple sensors onto the smart lens without obstructing vision or reducing comfort. Many researchers have utilized and optimized the space on the outer ring to combat this issue. Another method may include utilizing optical sensors that do not require electrical components, which will reduce the need to fit additional microprocessors. In addition, biosensors need to be capable of detecting low concentrations of biomarkers and small changes in IOP due to the subtle yet significant changes that can indicate the early stages of ocular diseases. Fluctuations in the body and ambient temperature can also affect the surface of the eye, hence leading to measurement errors and poor translation to clinical settings [84]. Furthermore, changes in temperature and humidity seem to affect IOP measurement when using SCLs [85]. Recent research by Li et al. (2024) examined how accurately IOP measurements were maintained amidst significant variations in ocular temperature and provided a solution in overcoming these limitations. The group developed an SCL that employs a dual inductor-capacitor-resistor system designed to compensate for temperature changes [86].

Thorough examination of regulations, approval procedures, and ethical considerations are also required for incoming and mostly unknown technologies such as SCLs. Currently, the Sensimed Triggerfish is the only FDA-approved device for IOP monitoring for glaucoma [87]. However, other companies such as GlakoLens are developing IOP monitoring SCLs, with six healthy volunteers involved in a pre-clinical pilot study [88]. Companies such as Sensimed and GlakoLens have become early pioneers in commercializing SCLs for widespread use. Securing regulatory approval by the FDA is the most critical step and thorough clinical trials and safety studies must be required to ensure compliance with various regulatory standards.

Furthermore, it is crucial for developers to have a clear understanding of their primary target market, such as ophthalmologists, and tailor the device characteristics to meet their specific needs. This includes ease of use, reliability under various conditions, accuracy and precision in data collection, price, and compatibility with existing diagnostic and treatment protocols. Furthermore, it important for SCL companies to ensure the device is covered by insurance providers to ensure widespread adoption by negotiating with healthcare payers and demonstrating the cost-effectiveness of the device in reducing healthcare costs overall. This involves devising a method for manufacturing the device at scale and maintaining good manufacturing practice protocols. Lastly, continuous and comprehensive training programs must be provided to healthcare providers to ensure proper usage. In fact, robust customer support allows for direct feedback, which prompts improvement to enhance functionality and user experience.

As with any wearable smart device, security and privacy are paramount in collecting and managing sensitive health data. Researchers must also encrypt transmitted data using resilient encryption standards to protect against unauthorized access. Additionally, patients should be informed about how data are collected, how and where they will be used, and who the information is shared with. This ensures transparency between the user and the provider, hence aligning with the principles of informed consent. Lastly, the storage of data must occur with complete data anonymity and reassurance that data will not be altered or tampered with. This helps to both protect patient privacy and complies with healthcare regulations, which mandates the safekeeping of personal health information.

## 5. Conclusions

As discussed in this review paper, SCLs will bring about a transformative advancement in the field of ophthalmology in both monitoring capabilities and therapeutic interventions. The innovative manufacturing of sensors, power supply, transmitters, and biomaterials allow for miniaturized circuitry to fit into the flexible, curved contact lens. By constantly monitoring the chemistry and physical properties of the eye in real time, SCLs have the potential to improve the management of chronic conditions such as glaucoma, diabetes, and dry eye disease. Furthermore, the prospects of SCLs extend beyond monitoring and biosensing, with potential applications in connecting with other wearable technologies. Seamless connection and communication with other health monitoring devices could enhance holistic health management. As SCL technology becomes more advanced, clearer regulatory, security, and ethical guidelines need to be implemented to help to protect potentially sensitive health data and address privacy concerns.

SCLs have a wide array of other uses as well. For example, Zhu et al. (2024) developed an SCL designed for eye tracking and wireless human–machine interaction. This has large implications in ophthalmology, as it can be used to enhance visual field testing, improve the detection of peripheral vision loss, and assist in the treatment of disorders like strabismus or amblyopia [80]. Tsai et al. (2024), developed an SCL display that could be utilized to implement augmented reality, opening the door to a wider range of applications such as navigation, AI assistance, or even providing real-time display data of health information [89]. Yao et al. (2023) developed an SCL for effective corneal injury repair [90], while Roostaei et al. (2022) investigated the use of SCLs for colour blindness correction [91].

Ultimately, SCLs are innovative technologies that have vast applications and may become an important instrument in the ophthalmologist’s toolkit for helping to diagnose, monitor, and treat ocular conditions.

## Figures and Tables

**Figure 1 micromachines-15-00856-f001:**
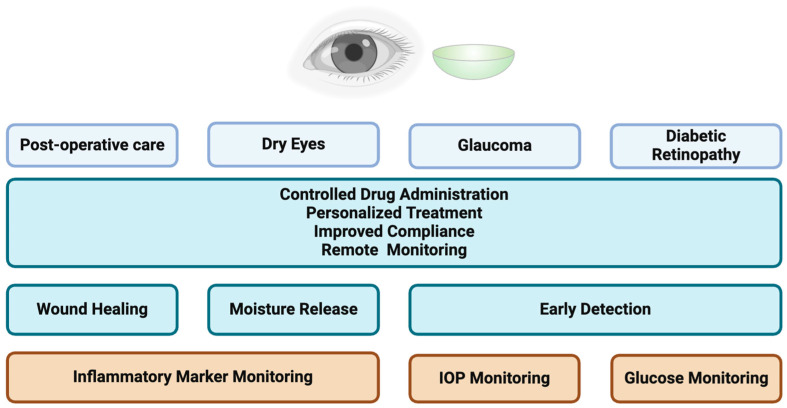
Therapeutic applications of smart contact lens (SCLs). SCLs can play an important role in the management and monitoring of patients in post-operative care, dry eyes, glaucoma, and diabetes. Made with permission from Biorender.

**Figure 2 micromachines-15-00856-f002:**
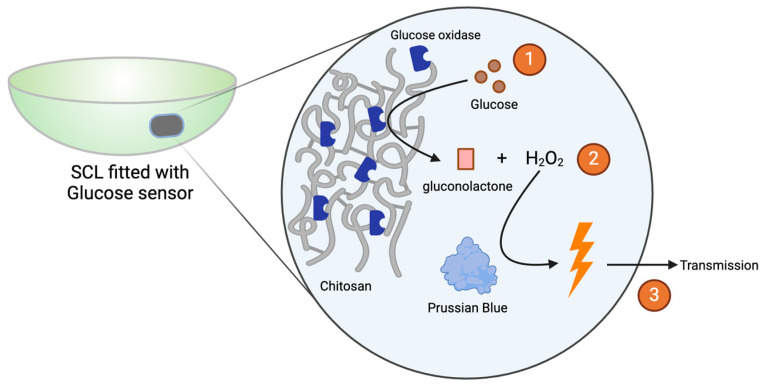
Simplified glucose sensor, as described by Park et al. (2024) [9]. Made with permission from Biorender.

## Data Availability

Not applicable.

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
