# Peer review of "Smart Contact Lenses in Ophthalmology: Innovations, Applications, and Future Prospects"

_micromachines, 2024, doi:10.3390/mi15070856_

Round 1

Reviewer 1 Report

Comments and Suggestions for Authors

The article:

  Smart Contact Lenses in Ophthalmology: Innovations, Applica-2 tions, and Future Prospects by:

Kevin Y. Wu, Archan Dave, Marjorie Carbonneau and Simon D. Tran,

deals with interesting approach the studies of  the  smart contact lenses represent a breakthrough intersection of medical science and innovative technology, offering transformative potential in ophthalmology. This is a review article in which the authors delve into the technological underpinnings of smart contact lenses, highlighting the current landscape and advances in biosensors, power sources, biomaterials, and ophthalmic information transmission. Additionally the authors in this review further applies new innovations to their emerging role in the diagnosis, monitoring, and management of various ocular conditions. In conclusion, the authors state that smart contact lenses are innovative technologies that have vast applications and may become an important instrument in an ophthalmologist’s toolkit for helping to diagnose, monitor, and treat ocular conditions.

Author Response

Dear Reviewer,

Thank you for your thoughtful and encouraging comments on our manuscript, "Smart Contact Lenses in Ophthalmology: Innovations, Applications, and Future Prospects." We appreciate the time and effort you have taken to review our work and provide valuable feedback.

We are pleased to hear that you found our approach to be interesting and recognized the transformative potential of smart contact lenses in ophthalmology. Your acknowledgment of our discussion on the technological underpinnings, advances in biosensors, power sources, biomaterials, and ophthalmic information transmission is greatly appreciated.

We are particularly grateful for your recognition of the innovations we highlighted and their emerging roles in the diagnosis, monitoring, and management of various ocular conditions. Your positive remarks reinforce the importance of this technology and its potential impact on ophthalmic practices.

We will take your feedback into account to further enhance the clarity and comprehensiveness of our review. Thank you once again for your insightful comments and for supporting our work.

Sincerely,

Reviewer 2 Report

Comments and Suggestions for Authors

There are two format errors in the text:

Line 170 on Page 4, a dot at the end of this paragraph is missing.

Line 453 on Page 10, the number of the section “4. Therapeutic Potential” should be “3. Therapeutic Potential”, as well as the numbers of the following sections need correcting.

Comments on the Quality of English Language

The manuscript entitled “Smart Contact Lenses in Ophthalmology: Innovations, Applications, and Future Prospects” is well written, and the authors systematically analyzed the recent development of smart contact lenses in ophthalmology, including five important aspects of SCLs, measurement of glucose and lactic acid, measurement of IOP, power supply, biomaterials and transmission of information. Therapeutic potential, challenges, and limitations were discussed and evaluated in the last part of the article, reasonably and convincingly.

Author Response

Thank you for your meticulous review and valuable suggestions regarding our manuscript, "Smart Contact Lenses in Ophthalmology: Innovations, Applications, and Future Prospects." We greatly appreciate your keen eye for detail and the constructive feedback you have provided.

Comments and Suggestions for Authors:

  1. We apologize for the formatting errors you identified:
    • Line 170 on Page 4: We will correct the missing dot at the end of the paragraph.
    • Line 453 on Page 10: We will adjust the section number "4. Therapeutic Potential" to "3. Therapeutic Potential," and correct the numbering of the subsequent sections accordingly.

Comments on the Quality of English Language: We are pleased to hear that you found the manuscript well-written and appreciated our systematic analysis of the recent developments in smart contact lenses in ophthalmology. Your positive feedback on our coverage of the five important aspects of SCLs—measurement of glucose and lactic acid, measurement of IOP, power supply, biomaterials, and transmission of information—is greatly valued. Additionally, your recognition of our discussion on the therapeutic potential, challenges, and limitations is deeply appreciated.

We will promptly address the formatting issues and ensure the manuscript is revised accordingly. Thank you once again for your thorough review and encouraging comments, which significantly contribute to the improvement of our work.

Sincerely,

Reviewer 3 Report

Comments and Suggestions for Authors

Smart Contact Lenses in Ophthalmology: Innovations, Applications, and Future Prospects

Comments:

This manuscript mainly discussed the technological underpinnings of smart contact lenses, emphasizing the current landscape and advancements in biosensors, power supply, biomaterials, and transmission of ocular information. This review synthesizes recent advances to provide an outlook on the state of smart contact lens technology. Furthermore, the authors discuss future directions, focusing on potential technological enhancements and new applications within ophthalmology.

1. Figure 1 is too simple and provides too little information. The author is advised to add some related eye diseases.

2. The author seems to have skipped the content of Chapter 3, directly moving from Chapter 2 to Chapter 4.

3. The subsections under Chapter 2 have too little direct relevance to each other, making the logical structure somewhat chaotic. It is recommended that the author create a separate chapter for each type of sensor.

4. The therapeutic applications in Chapter 4 can be categorised. It is recommended that the author summarise contact lenses aimed at the same type of disease together.

Comments on the Quality of English Language

Minor editing of English language required

Author Response

Dear Reviewer,

Thank you very much for your insightful comments and valuable suggestions. We appreciate the time and effort you have invested in reviewing our manuscript. We have carefully considered your feedback and have made the necessary revisions to address each of your comments. Please find below our responses to each point:

  1. Figure 1 is too simple and provides too little information. The author is advised to add some related eye diseases.

Thank you for pointing this out. We have enhanced Figure 1 to include additional information on related eye diseases. The updated figure now provides a more comprehensive overview that aligns with the discussed advancements in smart contact lens technology.

  1. The author seems to have skipped the content of Chapter 3, directly moving from Chapter 2 to Chapter 4.

We apologize for the numbering error in our manuscript. The section labeled as Chapter 4 should have indeed been Chapter 3. This has been corrected in the updated version of the manuscript to ensure proper sequential flow.

  1. The subsections under Chapter 2 have too little direct relevance to each other, making the logical structure somewhat chaotic. It is recommended that the author create a separate chapter for each type of sensor.

We appreciate your suggestion regarding the structure of Chapter 2. While we understand the rationale behind creating separate chapters for each type of sensor, we believe that discussing the technological underpinnings of smart contact lenses in a single chapter provides a more cohesive narrative. This approach allows us to present an integrated view of the technology, emphasizing how various sensors collectively contribute to the functionality of smart contact lenses. The following chapter focuses on the clinical applications, providing a clear delineation between technological and clinical aspects. We hope this explanation clarifies our rationale and provides a convincing argument for maintaining the current structure.

  1. The therapeutic applications in Chapter 4 can be categorized. It is recommended that the author summarize contact lenses aimed at the same type of disease together.

We agree with your suggestion and have revised Chapter 4 to categorize the therapeutic applications more clearly. The updated chapter now groups contact lenses aimed at similar types of diseases, making it easier for readers to understand the specific applications and benefits of smart contact lenses in various ophthalmic conditions.

Once again, we sincerely appreciate your thorough review and constructive feedback. We believe that the revisions have significantly improved the manuscript, and we hope that the updated version meets your expectations.

Thank you for your consideration.

Best regards,